# Pruritus in Palliative Care: A Narrative Review of Essential Oil-Based Strategies to Alleviate Cutaneous Discomfort

**DOI:** 10.3390/diseases13080232

**Published:** 2025-07-23

**Authors:** Sara Diogo Gonçalves

**Affiliations:** Clinical Academic Center of Trás-os-Montes and Alto Douro (CACTMAD), University of Trás-os-Montes and Alto Douro, 5000-801 Vila Real, Portugal; sgoncalves@utad.pt

**Keywords:** pruritus, palliative care, essential oils, aromatherapy, complementary therapy

## Abstract

Pruritus is a common and distressing symptom in palliative care, often resulting from complex underlying conditions such as cancer, chronic kidney disease, and liver failure. Conventional pharmacological treatments frequently offer limited relief and may produce undesirable side effects in this medically fragile population. Despite the high prevalence and impact of pruritus in palliative care, there is a lack of consolidated evidence on integrative non-pharmacological approaches. This narrative review explores the potential role of essential oils as a complementary approach to managing pruritus in palliative settings. A review of the literature was conducted to examine the mechanisms of action, safety considerations, and clinical outcomes associated with the use of essential oils, with a particular focus on their anti-inflammatory, neuromodulatory, and soothing properties. Evidence suggests that essential oils may provide symptom relief and enhance quality of life when integrated into multidisciplinary care; however, small sample sizes, heterogeneity, and methodological weaknesses often limit the findings of these studies. Furthermore, the long-term safety and antigenotoxic potential of essential oils remain underexplored. This narrative review concludes that while essential oils appear promising as adjunct therapies for pruritus, further rigorous research, particularly well-designed clinical trials and toxicological assessments, is needed to support their safe and effective use in palliative care.

## 1. Introduction

Pruritus, commonly referred to as itching, is a frequent and often underestimated symptom in palliative care, affecting patients with a wide range of advanced illnesses, including cancer, chronic kidney disease, and liver failure [1]. It is not only physically distressing but also significantly impairs sleep, mood, and overall quality of life. Despite its prevalence, pruritus remains challenging to treat effectively, particularly in the context of complex symptomatology and polypharmacy that characterizes palliative care [2].

In this narrative review, “palliative care” refers to specialized medical care that focuses on improving the quality of life for individuals living with serious, life-limiting illnesses. Palliative care encompasses symptom management, psychosocial support, and comfort care, irrespective of prognosis or treatment intent [3]. While palliative care is commonly associated with end-of-life care, it also applies to earlier phases of illness, where symptom relief is prioritized alongside disease-modifying treatments [4]. This review focuses on the role of essential oils in managing pruritus, specifically within the broader palliative framework, encompassing both advanced disease stages and end-of-life care, while acknowledging the complex symptom burden and high prevalence of cutaneous discomfort in these populations.

Conventional pharmacological therapies, such as antihistamines, corticosteroids, and gabapentinoids, provide limited relief in many cases and are often associated with undesirable side effects, such as sedation, cognitive impairment, immunosuppression, and increased fall risk, which are particularly problematic in frail or terminally ill patients [5,6,7]. As a result, there is growing interest in complementary and integrative approaches that can offer safe and holistic symptom management. Among these, aromatherapy with essential oils has gained attention due to its potential antipruritic, anti-inflammatory, and anxiolytic effects [8].

Conventional pharmacological therapies for pruritus—such as antihistamines, corticosteroids, gabapentinoids, opioid-receptor modulators, and certain antidepressants—can offer modest relief but are often limited by side effects (sedation, cognitive impairment, and immunosuppression) and inconsistent efficacy in neuropathic, cholestatic, or uremic itch. In palliative populations—frail, polypharmaceutical, and organ-impaired—the risk–benefit balance frequently favors exploring gentle, complementary modalities [6,9,10,11,12,13,14,15].

This narrative review aims to explore the current evidence surrounding the use of essential oils in the management of pruritus in palliative care, examining their mechanisms of action, clinical applications, and integration into multidisciplinary care practices. Specifically, the objectives are to: (1) describe the pathophysiology of pruritus in palliative populations; (2) examine the proposed mechanisms of action of essential oils, including modulation of inflammatory cytokines (e.g., IL-1β, IL-4, IL-31, and TNF-α), activation of transient receptor potential (TRP) ion channels such as TRPM8, and interactions with gamma-aminobutyric acid (GABA)-ergic signaling pathways; (3) summarize clinical applications and reported outcomes of essential oil-based therapies; and (4) discuss practical considerations for integration into multidisciplinary palliative care. These mechanisms may reduce itch perception, alleviate skin inflammation, and provide anxiolytic benefits. Particular attention is given to the relevance of individualized treatment approaches, safety considerations, and the need for further research, including antigenotoxicity studies, to support their responsible use in vulnerable populations.

## 2. Literature Search

This narrative review followed a rigorous methodological protocol, aiming to compile and critically analyze studies investigating the role of essential oils in stress control. A literature search was conducted across various scientific databases, including PubMed and Scopus, using search terms such as “pruritus,” “palliative care,” “essential oils,” “aromatherapy,” and “antipruritic.” Articles were included based on their relevance to the topic, with a focus on studies involving human subjects, palliative populations, or mechanisms related to essential oils and pruritus. Reviews, clinical trials, observational studies, and preclinical research were considered. Non-English articles and those unrelated to pruritus or essential oils were excluded.

## 3. Pathophysiology of Pruritus in Palliative Care

Pruritus, commonly described as an unpleasant sensation that provokes the desire to scratch, is a multifactorial symptom frequently encountered in palliative care settings [1]. According to a recent systematic review, pruritus is identified as the most common skin-related symptom among palliative care patients, surpassing other dermatological complaints such as xerosis, pressure ulcers, or drug-induced eruptions [16]. This highlights its clinical significance and underscores the need for targeted management approaches.

Its pathophysiology is complex and often involves overlapping mechanisms, including neuropathic, inflammatory, cholestatic, and uremic pathways. Understanding these mechanisms is crucial for developing effective, patient-centered strategies for symptom relief, particularly in populations where conventional therapies may be limited or contraindicated.

### 3.1. Mechanisms Involved in Pruritus

#### 3.1.1. Neuropathic Pruritus

Neuropathic pruritus arises from dysfunction or damage to the nervous system, either centrally or peripherally [17]. This form of itch is particularly relevant in cancer patients with spinal cord involvement, chemotherapy-induced neuropathy, or post-herpetic neuralgia. It is often resistant to antihistamines and may be accompanied by burning or tingling sensations, highlighting the importance of targeting neural pathways in treatment [18].

#### 3.1.2. Inflammatory Pruritus

Inflammation plays a crucial role in various dermatological and systemic conditions characterized by pruritus [19]. Cytokines such as interleukin (IL)-2, IL-4, IL-31, and tumor necrosis factor-alpha (TNF-α) have been implicated in mediating itch by activating sensory neurons. In palliative care, patients with skin metastases, xerosis, or atopic conditions may have inflammatory pathways that dominate the cause of itch [20,21].

#### 3.1.3. Cholestatic Pruritus

Pruritus is a hallmark symptom in patients with cholestasis, commonly seen in advanced liver disease or malignancies involving the hepatobiliary system. The pathogenesis is thought to involve the accumulation of bile salts, endogenous opioids, and lysophosphatidic acid (LPA), all of which may activate specific itch receptors or sensory nerve fibers. Pruritus in this context is often severe and generalized, significantly impairing rest and comfort [22].

#### 3.1.4. Uremic Pruritus

Patients with end-stage renal disease (ESRD), particularly those on dialysis, frequently experience uremic pruritus. Though the exact mechanism remains unclear, factors such as systemic inflammation, hyperparathyroidism, imbalanced calcium-phosphate metabolism, and accumulation of pruritogenic toxins are implicated. The itch is typically bilateral, symmetrical, and persistent, significantly contributing to distress and poor sleep [23,24].

#### 3.1.5. Other Contributing Factors

Additional contributors to pruritus in palliative care include xerosis (dry skin), drug reactions (e.g., opioids), and paraneoplastic syndromes. In many cases, the cause of pruritus is multifactorial, requiring a nuanced and individualized approach to management [1,25].

### 3.2. Common Underlying Conditions in Palliative Care

Pruritus can arise from a variety of life-limiting conditions commonly seen in palliative care:Cancer: Hematologic malignancies such as Hodgkin’s lymphoma are particularly associated with severe, unexplained pruritus. Solid tumors may also contribute indirectly via paraneoplastic syndromes, metastasis to the skin, or cholestasis from hepatic involvement [2].Liver Disease: Patients with primary biliary cholangitis, metastatic liver disease, or obstructive cholangiopathies frequently report intractable itch, often preceding other symptoms of hepatic dysfunction [26].Kidney Disease: In chronic kidney disease, particularly ESRD, pruritus can become a debilitating symptom that affects up to 40–50% of patients on dialysis [27].Neurological Disorders: Conditions such as multiple sclerosis, post-stroke syndromes, or spinal cord compression due to metastatic cancer can provoke neuropathic pruritus [28].HIV/AIDS: Immune dysregulation, opportunistic infections, and medication side effects contribute to a high prevalence of pruritus in terminal-stage HIV patients [29].

In addition to the major systemic conditions outlined above, several other factors can exacerbate pruritus in palliative care populations. These include age-related skin changes leading to xerosis, dermatological conditions such as eczema or psoriasis, medication-induced pruritus (particularly with opioids and certain antibiotics), and hormonal changes, including pregnancy-related pruritus [1,29,30]. The multifactorial nature of pruritus underscores the need for individualized management approaches in palliative care.

### 3.3. Impact on Quality of Life and Psychological Well-Being

Pruritus is increasingly recognized as a symptom with substantial impact on quality of life (QoL) and psychological well-being, particularly in palliative care populations. Beyond the immediate physical discomfort, persistent itching has been shown to impair sleep quality, concentration, social interaction, and emotional stability. Patients frequently report insomnia, irritability, and profound fatigue secondary to uncontrolled pruritus, contributing to overall functional decline [31,32].

Recent evidence highlights that pruritus severely impairs quality of life and psychological health across different patient groups, including those with chronic viral hepatitis and cholestatic liver conditions. Studies show a clear dose–response relationship between itch severity and QoL decline, with marked effects on sleep, emotional functioning, and social engagement [33]. Similarly, evidence in liver diseases shows pruritus correlates with significantly lower health-related quality of life scores, as demonstrated with particular detriments in domains related to emotional functioning, mental health, and sleep disturbance [34].

In dermatological conditions, recent analyses have shown that pruritus intensity is directly associated with impaired social life, mood disturbances, and decreased overall life satisfaction [35]. These findings are consistent with those presented by Jaworecka et al. [36], which confirm that itch severity strongly correlates with reduced quality of life scores and heightened psychological distress, particularly in cases of chronic and intractable itch.

In the palliative care context, these multidimensional effects of pruritus can further exacerbate suffering in individuals already facing complex symptom burdens. Addressing pruritus holistically, therefore, is not only a matter of controlling physical symptoms but also crucial for maintaining dignity, comfort, and psychological well-being in the end-of-life stage.

### 3.4. Diagnostic Considerations in Pruritus Assessment

The assessment of pruritus in palliative care relies primarily on clinical evaluation and patient-reported outcomes, as there are no definitive laboratory tests for pruritus itself [23]. Standard tools include the Numerical Rating Scale (NRS), where patients rate itch severity from 0 (no itch) to 10 (worst imaginable itch), and the 5-D Itch Scale, which evaluates Duration, Degree, Direction, Disability, and Distribution of pruritus. Additionally, the Visual Analog Scale (VAS) is used in some settings [37,38,39].

A comprehensive assessment also involves identifying aggravating factors (e.g., xerosis and medication use), reviewing dermatologic signs (e.g., excoriations and lichenification), and considering systemic causes through relevant medical history and laboratory investigations (e.g., liver and renal function tests, and full blood count). Accurate characterization of pruritus can guide the development of tailored interventions, including the appropriate use of complementary therapies, such as essential oils [40,41].

## 4. Essential Oil-Based Care in Palliative Care: Mechanisms, Applications, and Clinical Integration

We were motivated by the lack of review articles on essential oil-based care in palliative care. The use of essential oils (EOs) in supportive and palliative care has increased significantly, driven by growing patient interest in natural and integrative therapies [42,43]. Aromatherapy—defined as the therapeutic use of EOs derived from aromatic plants—has been explored as a non-pharmacological or pharmacological strategy to manage a variety of symptoms, including anxiety, dysphagia, pain, nausea, and pruritus [44]. Within the context of pruritus, EOs may offer gentle, holistic support that aligns with the ethos of palliative care: to relieve suffering, enhance comfort, and respect patient autonomy.

### 4.1. Introduction to Aromatherapy and Essential Oils

EOs are volatile compounds extracted from flowers, leaves, resins, bark, or roots of plants through methods such as steam distillation or cold pressing. Their pharmacological effects are attributed to complex mixtures of bioactive constituents, including terpenes, alcohols, esters, aldehydes, and phenols [45,46].

In aromatherapy, EOs can be used through several modalities—most commonly topical application, inhalation, or massage—to elicit physiological and psychological effects. Their use in pruritus management is supported by historical ethnobotanical practices and increasingly by emerging scientific evidence.

### 4.2. Proposed Mechanisms of Action for Antipruritic Effects

Several mechanisms have been proposed to explain the antipruritic properties of essential oils, many of which align with known pathophysiological pathways involved in pruritus:Anti-inflammatory Effects: Many EOs, including lavender (*Lavandula angustifolia*), chamomile (*Matricaria recutita*), and tea tree (*Melaleuca alternifolia*), have demonstrated anti-inflammatory activity. These actions are mediated through the inhibition of pro-inflammatory cytokines (e.g., IL-1β and TNF-α), reduction in oxidative stress, and suppression of cyclooxygenase pathways, which may reduce skin irritation and inflammation-driven itch [47,48,49].Soothing and Emollient Properties: EOs such as chamomile, rose, and sandalwood can improve skin barrier function, enhance hydration, and soothe damaged or xerotic skin, which may provide symptomatic relief in pruritus related to dryness or irritation [50,51,52].Neuromodulatory Effects: Certain EOs interact with neurotransmitter systems. For example, lavender has demonstrated affinity for GABA-A receptors, producing anxiolytic and sedative effects that may indirectly reduce itch perception. Menthol-containing EOs such as peppermint (*Mentha piperita*) can also modulate TRPM8 ion channels, providing a cooling sensation that may counteract the itch–scratch cycle [53,54].Antimicrobial and Skin-Protective Effects: Certain EOs possess antimicrobial properties that can help reduce secondary infections resulting from scratching, thereby preventing further skin damage and inflammation [55,56].

To better understand the therapeutic potential of EOs in managing pruritus, it is helpful to examine their specific bioactive components, mechanisms of action, and clinical relevance. Table 1 summarizes the key EOs most commonly associated with antipruritic effects, highlighting their active compounds, physiological targets, and potential indications in palliative care.

### 4.3. Review of the Evidence

While large-scale clinical trials remain limited, a growing body of evidence—comprising randomized controlled trials, pilot studies, and observational reports—supports the efficacy of essential oils in reducing the severity of pruritus in palliative and related populations.

A randomized controlled trial involving patients with chronic kidney disease found that the topical application of a 1.5% peppermint EO solution significantly reduced pruritus intensity compared to placebo, with minimal side effects [59].

In oncology settings, lavender and chamomile EOs have been used in massage or bath oils to reduce pruritus, improve sleep, and promote relaxation in terminally ill patients [57,58].

A pilot study on aromatherapy hand massage using lavender and bergamot reported reductions in subjective pruritus scores among elderly patients in long-term care [60,61].

Systematic and scoping reviews have concluded that while more rigorous research is needed, EOs may provide low-risk adjunctive support in managing pruritus, particularly when psychological stress or skin dryness are contributing factors [62,63,64].

Despite variability in study design and quality, the consistency of reported patient benefits across multiple populations supports the inclusion of EOs in multidisciplinary palliative care plans.

Taken together, these findings suggest that essential oils—particularly peppermint, lavender, and chamomile—can offer safe and effective adjunctive relief for pruritus, especially when psychological distress or skin inflammation contributes to symptom burden.

### 4.4. Safety Considerations and Routes of Application

EOs are potent bioactive substances, and safety must be prioritized, especially in the palliative care population, where skin integrity, organ function, and immune status may be compromised.

#### 4.4.1. Topical Application

This is the most common route used for pruritus. EOs should always be diluted in a carrier oil (e.g., sweet almond, jojoba, or fractionated coconut oil) to avoid skin irritation or sensitization. A typical safe dilution for adults in palliative care ranges from 0.5% to 2%, depending on the oil and individual sensitivity. The application may target localized areas or be integrated into full-body massage [65].

#### 4.4.2. Inhalation

Inhalation is a gentle and non-invasive method that can influence mood, perception of symptoms, and neurophysiological responses. Methods include using a diffuser, aroma stick, or placing drops on a cloth or pillow. This route may be preferable for patients with widespread pruritus or limited mobility [66].

#### 4.4.3. Oral Ingestion

Generally, not recommended in palliative care without expert supervision, due to the risk of toxicity, drug interactions, and inconsistent evidence regarding efficacy and safety [67,68].

#### 4.4.4. Precautions

Always consider allergies, sensitivities, and potential interactions with existing medications.Avoid known dermal irritants (e.g., cinnamon, clove, and oregano) on broken or sensitive skin.EOs should not be used on mucosal areas or open wounds unless under specialist guidance.

EOs, when integrated thoughtfully into care plans, offer a low-risk, potentially effective adjunctive therapy for pruritus in palliative care. Their multimodal mechanisms—combining physiological relief with emotional and sensory support—make them particularly well-suited for this vulnerable population.

### 4.5. Clinical and Ethical Considerations for Use of Essential Oils in Pruritus Management

The integration of EOs into palliative care settings must be both evidence-informed and person-centered. While the scientific evidence for their efficacy in pruritus relief continues to develop, clinical experience and patient-reported outcomes suggest that EOs may offer meaningful improvements in comfort, dignity, and well-being. Their gentle nature, multimodal mechanisms, and alignment with holistic care principles make them particularly well-suited to the palliative context.

Successful use of EOs in pruritus management requires thoughtful consideration of several clinical and practical factors:

#### 4.5.1. Patient Preferences and Values

In palliative care, respecting the patient’s beliefs, preferences, and autonomy is paramount [69]. EOs may appeal to patients who prefer natural or non-pharmacological options, have had prior positive experiences with aromatherapy, or are reluctant to take additional medications due to pill burden or side effects. Engaging patients in shared decision-making can improve satisfaction and adherence to integrative interventions [70].

#### 4.5.2. Safety and Contraindications

Despite their natural origin, EOs are potent and must be used with care [71]. Particular attention should be paid to:Skin integrity: Avoid use on broken, fragile, or highly inflamed skin.Sensitivities and allergies: Patch testing may be advisable before topical application.Organ function: Patients with hepatic or renal impairment may have altered metabolism or excretion of oil constituents.Potential drug interactions: Though rare, interactions with sedatives, anticoagulants, or hormone-sensitive conditions may occur with some oils (e.g., fennel and clary sage).Mode of application: Topical use should involve proper dilution (typically 0.5–2% in a carrier oil), while inhalation should be conducted in a well-ventilated environment to avoid overwhelming fragrance [72].

#### 4.5.3. Ethical and Cultural Sensitivity

Some individuals may object to scents or treatments perceived as non-medical. Staff must be sensitive to cultural norms and personal boundaries, and offer alternatives where appropriate [73,74].

### 4.6. Role of Multidisciplinary Teams

The integration of aromatherapy into clinical care is most effective when coordinated by a multidisciplinary team, combining clinical oversight with therapeutic expertise.

Nurses play a central role in identifying symptoms, applying topical treatments, educating patients, and monitoring responses to aromatherapy interventions. Their frequent contact with patients makes them ideal advocates for integrative care.

Aromatherapists, particularly those with training in clinical or medical aromatherapy, can provide in-depth knowledge about oil selection, safety, and formulation. Collaboration with licensed aromatherapists ensures that essential oil use is informed by evidence and tailored to each patient’s specific needs and vulnerabilities.

Palliative care physicians provide clinical oversight, ensuring that the use of essential oils does not interfere with pharmacological therapies or exacerbate existing conditions. Their support is essential in integrating aromatherapy safely into care plans, especially when managing complex symptoms or advanced disease.

Pharmacists may also contribute by assessing possible interactions or advising on the pharmacokinetics of essential oil constituents.

This collaborative approach ensures that the integration of essential oils enhances care without compromising safety or clinical standards [57,65].

### 4.7. Practical Guidelines and Protocols

Although formal guidelines for the use of essential oils in pruritus are still limited, several institutions and professional organizations have begun to incorporate aromatherapy into their palliative care protocols. Common practices include (Table 2)

Assessment and documentation of pruritus characteristics, severity (e.g., using a Visual Analog Scale or 5-D Itch Scale), triggers, and patient history [37]. The 5-D Itch Scale is a validated tool designed to assess five dimensions of pruritus: Duration (frequency of itching episodes), Degree (severity), Direction (change over time), Disability (impact on daily activities such as sleep, leisure, housework, and work/school), and Distribution (affected body areas). It provides a more comprehensive evaluation compared to unidimensional scales, such as the NRS or VAS, offering valuable insights into both symptom severity and functional impact [37]. The scale is particularly useful in chronic and complex cases of pruritus, allowing clinicians to monitor therapeutic responses over time.Trial interventions starting with a patch test and observation of response to a low-dose topical or inhaled preparation [75].Standardized dilution protocols, typically:
○0.5–1% for frail, elderly, or terminally ill patients.○1–2% for more robust individuals with intact skin [65].Choice of carrier oils such as jojoba, sweet almond, or fractionated coconut oil, depending on skin condition and allergies.Clear documentation of EOs used, dose, route of application, frequency, and patient feedback.

Some hospice and integrative care centers have developed internal protocols that include staff training, safety data sheets, and informed consent procedures. Resources such as the International Federation of Professional Aromatherapists (IFPA) and National Association for Holistic Aromatherapy (NAHA) provide best practice guidelines and training pathways for integrating aromatherapy into healthcare.

In summary, the thoughtful integration of essential oils into pruritus management in palliative care can enhance quality of life, reduce symptom burden, and support the emotional and sensory needs of patients. Multidisciplinary collaboration, safety awareness, and person-centered implementation are essential for ensuring therapeutic benefit and upholding the values of compassionate care.

## 5. Discussion

The present review highlights the multifactorial nature of pruritus in palliative care and underscores the potential role of EOs as a complementary strategy to mitigate cutaneous discomfort. While pharmacological treatments remain a mainstay, their limitations—including side effects, diminished tolerability, and reduced efficacy in certain types of itch—open the door for integrative interventions that are safer, gentler, and more aligned with the principles of holistic palliative care.

The evidence suggests that EOs, through their anti-inflammatory, neuromodulatory, and skin-soothing properties, may offer meaningful symptom relief, especially when tailored to the individual’s type of pruritus and overall clinical context [76]. Their potential to contribute to improved sleep, reduced anxiety, and enhanced emotional well-being further strengthens the rationale for their inclusion in multidisciplinary care plans.

However, despite promising preliminary findings, several gaps and limitations must be addressed before essential oils can be fully endorsed as part of routine pruritus management protocols in palliative care.

### 5.1. Clinical Implications

The evidence reviewed suggests that essential oils—particularly peppermint, lavender, chamomile, and bergamot—may offer a meaningful adjunct to conventional pruritus management in palliative care. Their multimodal actions, including anti-inflammatory, antipruritic, anxiolytic, and skin-soothing effects, align well with the multifactorial nature of pruritus in advanced illness.

In clinical practice, essential oils can be incorporated via massage, compresses, baths, or diffusion, offering both localized relief and psychospiritual comfort. For example, the topical application of peppermint oil may be beneficial for uremic or neuropathic itch, while lavender and chamomile oils may be better suited for patients with concurrent anxiety or sleep disturbances. These interventions are generally well tolerated, low cost, and compatible with home or hospice settings.

Importantly, clinicians should consider individual preferences, skin integrity, comorbidities, and potential contraindications, such as allergies, broken skin, or phototoxicity (e.g., with non-FCF bergamot oil). The integration of aromatherapy into multidisciplinary care requires collaboration with trained professionals to ensure safety and appropriate dosing.

While further high-quality trials are needed, the current evidence supports the cautious and individualized use of essential oils as part of a holistic strategy for alleviating pruritus in palliative care contexts.

### 5.2. Limitations

Several limitations of this review must be acknowledged. The literature available on the use of EOs for pruritus in palliative care is heterogeneous, encompassing studies with varied methodologies, small sample sizes, and inconsistent outcome measures. A significant proportion of these studies were not explicitly conducted in palliative populations but rather in related contexts such as dermatology, nephrology, or geriatrics. As a result, extrapolation of findings to end-of-life settings must be approached with caution.

Moreover, many of the included studies lack control groups or standardized protocols for EO selection, dilution, and application methods, introducing a degree of variability that limits reproducibility. The absence of large-scale randomized controlled trials reduces the strength of the evidence, and publication bias remains a concern, particularly in the field of complementary therapies, where positive results may be overrepresented. Another important limitation is the scarcity of mechanistic and toxicological data. Although proposed pathways for antipruritic effects are biologically plausible, they are often inferred from in vitro or animal models and lack direct confirmation in human clinical settings, especially in palliative care. This gap is particularly relevant when considering long-term use, possible interactions with conventional medications, and the unique physiological vulnerabilities of terminally ill patients.

### 5.3. Future Research

In light of the limitations discussed, future research must aim to produce higher-quality, clinically relevant evidence. Before large-scale implementation, preliminary studies are needed to establish the safety and potential efficacy of essential oils in this context. Once this foundational evidence is available, well-designed randomized controlled trials targeting pruritus in palliative care populations will be essential. These trials should employ validated instruments for pruritus assessment and clearly defined protocols for the use of essential oils, including dosage, route of administration, and treatment duration.

A crucial consideration for future research is characterizing pruritus by its underlying cause. Given the differing pathophysiological mechanisms of neuropathic, cholestatic, uremic, and inflammatory itch, it is likely that certain essential oils may be more effective for specific subtypes. Tailoring aromatherapeutic interventions based on the nature of pruritus could improve therapeutic precision and patient outcomes.

Equally critical is the need to deepen our understanding of the safety profile of EOs, particularly when used chronically or in combination with other palliative treatments. Antigenotoxicity testing represents a key area for further investigation [55,77]. Patients receiving palliative care often face cumulative genotoxic stress from their primary disease and its treatments, such as chemotherapy or radiotherapy. Therefore, it is essential to ensure that the addition of natural products does not contribute to further genetic instability. Future studies should explore the genotoxic and antigenotoxic properties of EOs and their major constituents through in vitro, in vivo, and clinical pharmacovigilance approaches.

Finally, the integration of modern molecular tools, such as omics technologies and systems biology, could provide new insights into the bioactivity and interaction networks of essential oils. These approaches could help identify biomarkers of efficacy or toxicity, support personalized aromatherapy protocols, and bridge the gap between empirical use and evidence-based clinical application. In doing so, essential oils may transition from being perceived as complementary to becoming validated, integrative components of symptom management in palliative care.

## 6. Conclusions

Pruritus is a common and burdensome symptom in palliative care, often resistant to conventional treatments. This review highlights the potential of essential oils as a complementary approach, offering antipruritic effects through anti-inflammatory and neuromodulatory mechanisms. While early evidence is promising, further research—particularly focused on safety, standardization, and antigenotoxicity—is essential. When integrated thoughtfully by multidisciplinary teams, essential oils may enhance comfort and quality of life for patients in palliative settings.

## Figures and Tables

**Table 1 diseases-13-00232-t001:** Proposed Mechanisms of Action for Selected Essential Oils in Pruritus Relief.

Essential Oil	Key Active Compounds	Mechanisms of Action	Target Pathways/Effects	Indications in Pruritus	Efficacy Evidence
Lavender (*Lavandula angustifolia*)	Linalool, linalyl acetate	Neuromodulation via GABA-A receptor modulation; anti-inflammatory; anxiolytic	CNS calming, decreased itch perception, reduced inflammation	Neuropathic itch, anxiety-related itch, generalized pruritus	Aromatherapy massage ↓ pruritus and improved sleep in cancer patients [57]
Chamomile (*Matricaria recutita*/*Chamaemelum nobile*)	α-bisabolol, chamazulene	Inhibition of pro-inflammatory cytokines (IL-1β, TNF-α); antioxidant; skin barrier repair	Anti-inflammatory, barrier support	Inflammatory dermatoses, dry-skin-related itch, atopic-prone skin	Bath therapy ↓ skin irritation in pediatric eczema [58]
Peppermint (*Mentha piperita*)	Menthol, menthone	TRPM8 ion channel activation (cooling sensation); local analgesic effect	Sensory nerve modulation, anti-itch signaling	Uremic pruritus, cholestatic pruritus, thermal/irritant pruritus	RCT in CKD patients: ↓ itch intensity with 1.5% solution vs. placebo [59]
Tea Tree (*Melaleuca alternifolia*)	Terpinen-4-ol, α-terpineol	Antimicrobial; inhibits prostaglandins and histamine release	Skin infection control, inflammation reduction	Secondary infection from scratching, contact dermatitis	Preclinical studies show ↓ histamine-induced itching [56]
Sandalwood (*Santalum album*)	Santalol	Soothing, emollient, and anti-inflammatory activity	Skin hydration, TNF-α modulation	Xerotic pruritus, chronic inflammatory pruritus	Preclinical and dermatological use shows anti-inflammatory and skin-soothing effects in xerosis [50]
Geranium (*Pelargonium graveolens*)	Citronellol, geraniol	Antioxidant and mild anti-inflammatory properties	Oxidative stress reduction, peripheral sensory modulation	Stress-related or multifactorial pruritus	No direct RCT; antioxidant and sensory-modulating properties suggest potential benefit in stress-related pruritus
Bergamot (*Citrus bergamia*) ^†^	Limonene, linalool, bergapten (furocoumarin) ^†^	Anxiolytic via serotonergic pathways; potential phototoxicity	Mood enhancement, CNS neuromodulation	Psychological pruritus, anxiety-exacerbated itch	In combination with lavender, showed reduced pruritus and improved mood in elderly/palliative care patients [60]

^†^*:* Bergamot essential oil should be used in **furocoumarin-free (FCF)** form to avoid phototoxic reactions when applied topically; ↓: decrease; CKD: chronic kidney disease; CNS: central nervous system; GABA: gamma-aminobutyric acid; IL: interleukin; RCT: randomized clinical trial; TNF: tumor necrosis factor; TRPM: transient receptor potential melastatin.

**Table 2 diseases-13-00232-t002:** Summary of Aromatherapy Guidelines for Pruritus in Palliative Care.

Aspect	Details
Recommended EOs	Lavender (*Lavandula angustifolia*), Roman chamomile (*Chamaemelum nobile*), Peppermint (*Mentha × piperita*), Frankincense (*Boswellia carterii*), Tea tree (*Melaleuca alternifolia*)
Dilution Ratios	- 0.5–1% for frail or elderly patients - 1–2% for robust patients with intact skin
Routes of Administration	- Topical (localized application, massage) - Inhalation (diffuser, personal inhaler)
Carrier Oils	Jojoba, Sweet almond, Fractionated coconut oil
Contraindications	- Open wounds, active skin infections - Known EO allergies or sensitivities - Use caution in immunocompromised patients or those with respiratory conditions

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
