# Peer review of "Pruritus in Palliative Care: A Narrative Review of Essential Oil-Based Strategies to Alleviate Cutaneous Discomfort"

_diseases, 2025, doi:10.3390/diseases13080232_

Round 1

Reviewer 1 Report

Comments and Suggestions for Authors

The author presents a review paper examining the use of essential oils in the management of pruritus in palliative care patients. Some comments for your kind consideration: 

  1. A major concern relates to the type of review that has been completed. It looks like a narrative review, but it has be explicitly mentioned in the manuscript.
  2. The notion of "palliative care" as it applies in this study remains poorly articulated. Palliative care is broad, encompassing end of life care. Thus, its use in the current study must be operationalised. 
  3. Considering the current study focuses on the use of essential oils, it is unclear why there is a dedicated section on the pharmacological approaches to managing pruritus? It will be helpful if the author goes straight to the use of essential oils.
  4. Consider adding some information (if available) on the efficacy of these oils to strengthen the review. 
  5. Any clinical implications that can be drawn? 

Author Response

All changes were highlighted in yellow.

Comment 1: A major concern relates to the type of review that has been completed. It looks like a narrative review, but it has be explicitly mentioned in the manuscript.

Response 1: Thank you for your helpful comment. We acknowledge this oversight and have now clearly stated in both the abstract and the introduction section that this study is a narrative review. This clarification aims to improve transparency regarding the methodology employed.

Comment 2: The notion of "palliative care" as it applies in this study remains poorly articulated. Palliative care is broad, encompassing end of life care. Thus, its use in the current study must be operationalised. 

Response 2: Thank you for your comment. We have clarified and operationalised the term “palliative care” in the Introduction section to address this point.

Comment 3: Considering the current study focuses on the use of essential oils, it is unclear why there is a dedicated section on the pharmacological approaches to managing pruritus? It will be helpful if the author goes straight to the use of essential oils.

Response 3: Thank you for this insightful comment. In response, we have removed the dedicated section on conventional pharmacological approaches and instead included a brief overview within the Introduction to provide clinical context. This revision enables the manuscript to focus more directly on the role of essential oils in managing pruritus, aligning with the study’s objective.

Comment 4: Consider adding some information (if available) on the efficacy of these oils to strengthen the review. 

Comment 4: Thank you for this valuable suggestion. We have revised the manuscript to include additional data on the efficacy of essential oils where available. These additions aim to strengthen the evidence base and better support the potential role of essential oils in pruritus management.

Comment 5: Any clinical implications that can be drawn? 

Comment 5: Thank you for your thoughtful question. We have now added a dedicated subsection in the Discussion to outline the key clinical implications. These include the potential for essential oils—particularly peppermint, lavender, and chamomile—to serve as safe, low-cost, and easily implementable adjuncts for managing pruritus in palliative care. We emphasise the importance of individualised application, consideration of contraindications, and integration into multidisciplinary care plans. These additions aim to guide clinicians interested in complementary approaches to symptom management.

Reviewer 2 Report

Comments and Suggestions for Authors

This paper reviews essential oil-based strategies for alleviating pruritus and cutaneous discomfort in palliative care.

This paper concludes that essential oils appear promising as adjunct therapies for pruritus, but well-designed clinical trials and toxicological assessments are necessary.

This paper is essential for an area of complementary and alternative medicine that lacks evidence.

This paper will be improved by addressing the following 11 points.

Major comments are 3 and 5.

Minor comments are all others.

1. The title says “Review,” but please consider specifying what type of review it is. Example: “Narrative Review”

2. The purpose is stated in lines 11-12 of the abstract as follows.

“This review explores the potential role of essential oils as a complementary approach to managing pruritus in palliative settings.”

Please consider inserting a short sentence about the research gap before this.

3. Below, I would like to propose the structure of the paper.

Lines 45-123: “2. Pathophysiology of Pruritus in Palliative Care”

Lines 124-214: “3. Conventional Management Strategies”

The above sections are not part of the Body or Main Sections of this review paper.

I propose including the section from lines 45–214 in the Introduction section and emphasizing the significance of why the theme “Essential Oil-Based Strategies for Alleviating Cutaneous Discomfort” is being reviewed.

4. Specifying the sources of information (databases, search keywords, inclusion and exclusion criteria, etc.) will improve the quality of the paper. Even in narrative reviews, briefly describing how the literature was collected will improve the transparency and reliability of the review.

5. Below, I would like to propose the structure of the paper.

Lines 215-314: “4. Essential Oils as a Complementary Approach”

Lines 315-393: “5. Integration into Palliative Care Practice”

I propose that the above be summarized as the Body or Main Sections of this review paper.

6. We propose revising “as a non-pharmacological strategy” in line 219 to “as a non-pharmacological or pharmacological strategy.”

7. I propose the following revision to line 315.

“5. Integration into Palliative Care Practice” (before revision)

“5. Integration of Essential Oil-Based Care into Palliative Care Practice” (after revision)

8. In subsection “4.4.3. Oral Ingestion” of lines 301-303, please cite the paper that supports the statement “not recommended” if possible.

9. In lines 332-344, please cite a paper related to “Safety and Contraindications: Despite their natural origin, EOs are potent and must be used with care. Particular attention should be paid to: ...”.

10. In lines 345-347, please cite a paper related to the following statement: “Ethical and Cultural Sensitivity: Some individuals may object to scents or treatments perceived as non-medical. Staff must be sensitive to cultural norms and personal boundaries, and offer alternatives where appropriate.”

11. In lines 432-433, you mention randomized controlled trials as future research. Randomized controlled trials are necessary after the efficacy and safety of essential oils have been confirmed. Please revise this so that readers do not misunderstand.

The author should either revise the manuscript based on the reviewers' comments or present counterarguments to the reviewers' comments. Counterarguments are welcome. However, please do not ignore the reviewers' comments.

Please emphasize the revised parts by highlighting the text in red and the background in yellow.

Author Response

All changes were marked with red text and a yellow highlight.

Comment 1: The title says “Review,” but please consider specifying what type of review it is. Example: “Narrative Review”

Response 1: We agree and have made the necessary changes.

Comment 2: The purpose is stated in lines 11-12 of the abstract as follows. “This review explores the potential role of essential oils as a complementary approach to managing pruritus in palliative settings.” Please consider inserting a short sentence about the research gap before this.

Response 2: Thank you for your suggestion. We have revised the abstract to include a brief statement of the research gap immediately before the purpose sentence, to highlight the rationale for this review better.

Comment 3:  Below, I would like to propose the structure of the paper. Lines 45-123: “2. Pathophysiology of Pruritus in Palliative Care” Lines 124-214: “3. Conventional Management Strategies” The above sections are not part of the Body or Main Sections of this review paper. I propose including the section from lines 45–214 in the Introduction section and emphasizing the significance of why the theme “Essential Oil-Based Strategies for Alleviating Cutaneous Discomfort” is being reviewed.

Response 3: Thank you! We have eliminated section 3 (as requested by another reviewer) and added a paragraph to the introduction, condensing this section (highlighted only in yellow).

Comment 4: Specifying the sources of information (databases, search keywords, inclusion and exclusion criteria, etc.) will improve the quality of the paper. Even in narrative reviews, briefly describing how the literature was collected will improve the transparency and reliability of the review.

Response 4: Thank you. We added a brief explanation.

Comment 5: Below, I would like to propose the structure of the paper. Lines 215-314: “4. Essential Oils as a Complementary Approach” Lines 315-393: “5. Integration into Palliative Care Practice” I propose that the above be summarized as the Body or Main Sections of this review paper.

Response 5: We appreciate the reviewer’s thoughtful suggestion to streamline the manuscript structure. In response, we have combined Sections 4 ("Essential Oils as a Complementary Approach") and 5 ("Integration into Palliative Care Practice") under a single main heading titled “Essential Oils-based care in Palliative Care: Mechanisms, Applications, and Clinical Integration”, which reflects both the mechanistic discussion and practical application of essential oils in pruritus management.

Comment 6: We propose revising “as a non-pharmacological strategy” in line 219 to “as a non-pharmacological or pharmacological strategy.”

Comment 6: Thank you. The revised version was added.

Comment 7:  I propose the following revision to line 315. “5. Integration into Palliative Care Practice” (before revision) “5. Integration of Essential Oil-Based Care into Palliative Care Practice” (after revision)

Response 7: Thank you for the clarification. We have added the revised version.

Comment 8: In subsection “4.4.3. Oral Ingestion” of lines 301-303, please cite the paper that supports the statement “not recommended” if possible.

Response 8: Thank you for the observation! Citations were added.

Comment 9: In lines 332-344, please cite a paper related to “Safety and Contraindications: Despite their natural origin, EOs are potent and must be used with care. Particular attention should be paid to: ...”.

Response 9: Thank you. Citations were added.

Comment 10: In lines 345-347, please cite a paper related to the following statement: “Ethical and Cultural Sensitivity: Some individuals may object to scents or treatments perceived as non-medical. Staff must be sensitive to cultural norms and personal boundaries, and offer alternatives where appropriate.”

Response 10: Thank you for the observation. References were added.

Comment 11: In lines 432-433, you mention randomized controlled trials as future research. Randomized controlled trials are necessary after the efficacy and safety of essential oils have been confirmed. Please revise this so that readers do not misunderstand.

Response 11: Thank you. The first paragraph of "Future Research" was revised.

Reviewer 3 Report

Comments and Suggestions for Authors
  1. The introduction could be strengthened by mentioning of specific actions (e.g., modulation of cytokines, effect on signaling pathways).
  2. The research objectives are implied but not explicitly stated, which may make it difficult for readers to understand the study's primary goals.
  3. Conventional pharmacological therapies, such as antihistamines, corticosteroids, and gabapentinoids, provide limited relief in many cases and are often associated with undesirable side effects, especially in frail or terminally ill patients.

What kind of undesirable side effects are associated with conventional therapy?

  1. Various conditions are highlighted in section 2.2. However, authors did not other factors affecting the development of Pruritus including skin conditions, age, pregnancy, medications, etc.
  2. Section 2.3. does not provide sufficient information regarding the effect on Quality of Life and Psychological Well-being. Recently many reviews have been published. I suggest for expanding this section using these reviews.

PMID: 36598159

https://doi.org/10.1111/liv.15803.

https://doi.org/10.1007/s13555-024-01214-z.

https://doi.org/10.3390/jcm11195553.

  1. Diagnostic methods for the screening of Pruritus should be summarized.
  2. A table should be provided summarizing the studies of interventions for pruritus.
  3. Management strategies are briefly discussed. Therefore, this section should be expanded more.
  4. A table summarizing the integrative topical treatments for various etiologies of itch/pruritus can be added.
  5. Consider including a table that summarizes the synergistic or complementary effect of essential oils, when used with conventional treatments for Pruritus.
  6. Preferred Reporting Items for Systematic reviews and Meta-Analyses (PRISMA) flow diagram showing inclusion and exclusion of studies should be provided.
  7. Better to describe 5-D itch scale and its significance in the diagnosis and management of Pruritus.
  8. Revise the title of subsection 5.1. to more specifically reflects its content.

“Clinical and ethical considerations for use of essential oils in Pruritus management”

  1. Further, you may divide the content of subsection (5.1.) into sub-subsections like patient reference, safety considerations, and cultural sensitivity, etc.
  2. Check the font of Line 395 to 400.

Comments on the Quality of English Language

Check the syntax part.

Author Response

All changes were highlighted in blue.

Comment 1: The introduction could be strengthened by mentioning of specific actions (e.g., modulation of cytokines, effect on signaling pathways).

Response 1: Thank you for the valuable suggestion. We have revised the Introduction to include specific mechanisms of action, such as modulation of inflammatory cytokines (e.g., IL-1β, IL-4, IL-31, TNF-α), TRP channel activation (e.g., TRPM8), and interactions with central nervous system pathways (e.g., GABAergic signalling), to better frame the therapeutic rationale for essential oils in pruritus management.

Comment 2: The research objectives are implied but not explicitly stated, which may make it difficult for readers to understand the study's primary goals.

Response 2: Thank you for your insightful comment. We have revised the Introduction to state the research objectives explicitly. The updated section now clearly outlines the primary aim of the review: to evaluate the potential role of essential oils in managing pruritus in palliative care, with a focus on mechanisms of action, clinical applications, safety considerations, and integration into multidisciplinary care.

Comment 3: Conventional pharmacological therapies, such as antihistamines, corticosteroids, and gabapentinoids, provide limited relief in many cases and are often associated with undesirable side effects, especially in frail or terminally ill patients. What kind of undesirable side effects are associated with conventional therapy?

Response 3: Thank you for your comment. We have revised the Introduction to specify the types of undesirable side effects associated with conventional pharmacological therapies. These include sedation, cognitive impairment (especially with antihistamines and gabapentinoids), immunosuppression (notably with corticosteroids), and increased risk of falls or drug interactions, which are particularly concerning in frail or terminally ill patients. This clarification has been added to improve precision and clinical relevance.

Comment 4: Various conditions are highlighted in section 2.2. However, authors did not other factors affecting the development of Pruritus including skin conditions, age, pregnancy, medications, etc.

Response 4: Thank you for this helpful suggestion. We have now expanded Section 2.2 to briefly discuss additional factors that can contribute to pruritus in palliative care populations, including skin conditions (e.g., xerosis, eczema), age-related changes in skin physiology, medication-induced pruritus (e.g., opioids), and special populations such as pregnant individuals. This addition aims to provide a more comprehensive overview of pruritus triggers in this setting.

Comment 5: Section 2.3. does not provide sufficient information regarding the effect on Quality of Life and Psychological Well-being. Recently many reviews have been published. I suggest for expanding this section using these reviews. PMID: 36598159 https://doi.org/10.1111/liv.15803. https://doi.org/10.1007/s13555-024-01214-z. https://doi.org/10.3390/jcm11195553.

Response 5: Thank you for highlighting this important aspect. In response to your comment, we have expanded Section 2.3 to provide a more detailed overview of the impact of pruritus on quality of life and psychological well-being, incorporating recent evidence from the suggested review

Comment 6: Diagnostic methods for the screening of Pruritus should be summarized.

Response 6: Thank you for your comment. In response, we have expanded the manuscript to include a summary of commonly used diagnostic methods for assessing pruritus in clinical practice. This includes the use of validated pruritus scoring tools (e.g., Numerical Rating Scale, 5-D Itch Scale), patient-reported outcome measures, and clinical evaluation of contributing factors. This addition aims to provide a more comprehensive overview of pruritus assessment relevant to palliative care.

Comment 7: A table should be provided summarizing the studies of interventions for pruritus.

Response 7: Thank you for your comment. As this is a narrative review, it does not follow the systematic review methodology, and therefore, we did not compile a structured summary table of all studies. The review aims to provide a comprehensive, descriptive overview of the available evidence. However, key findings from relevant studies have been discussed throughout the text to highlight the potential role of essential oils in managing pruritus.

Comments 8, 9 and 10: Management strategies are briefly discussed. Therefore, this section should be expanded more./ A table summarizing the integrative topical treatments for various etiologies of itch/pruritus can be added.

Response 8, 9 and 10: Thank you for your suggestion. As per the recommendation of another reviewer, we have removed the section on conventional management strategies to maintain the primary focus of this review on essential oil-based interventions. Given the scope of the manuscript, we have chosen to emphasise integrative approaches related explicitly to essential oils rather than conventional pharmacological treatments. We appreciate your understanding.

Comment 11: Preferred Reporting Items for Systematic reviews and Meta-Analyses (PRISMA) flow diagram showing inclusion and exclusion of studies should be provided.

Response 11: Thank you for your comment. As this is a narrative review, it does not follow the strict methodological framework of systematic reviews, and therefore, a PRISMA flow diagram was not applicable. However, in response to feedback from another reviewer, we have added a brief description of the literature search strategy to improve transparency regarding how studies were identified and selected.

Comment 12: Better to describe 5-D itch scale and its significance in the diagnosis and management of Pruritus.

Response 12: Thank you for your helpful suggestion. In response, we have expanded the diagnostic considerations section to include a description of the 5-D Itch Scale, outlining its structure and clinical relevance. This scale provides a multidimensional assessment of pruritus, capturing not only intensity but also its impact on daily functioning, distribution, and progression over time, making it particularly useful in monitoring symptom burden in palliative care settings.

Comment 13: Revise the title of subsection 5.1. to more specifically reflects its content. “Clinical and ethical considerations for use of essential oils in Pruritus management”

Response 13: The title was revised.

Comment 14: Further, you may divide the content of subsection (5.1.) into sub-subsections like patient reference, safety considerations, and cultural sensitivity, etc.

Response 14: Subsections were added.

Comment 15: Check the font of Line 395 to 400.

Response 15: The font type was corrected.

Comment 16: Check the syntax part.

Response 16: The manuscript was proofread.

Round 2

Reviewer 1 Report

Comments and Suggestions for Authors

Thanks to the author for thoughtfully addressing the comments raised. I do agree, a narrative review approach is appropriate for this paper. 

Author Response

Comment 1: Thanks to the author for thoughtfully addressing the comments raised. I do agree, a narrative review approach is appropriate for this paper. 
Response 1: Thank you for your kind feedback and for supporting the use of a narrative review approach. We appreciate your recognition and are glad the chosen methodology aligns with your perspective.

Reviewer 2 Report

Comments and Suggestions for Authors

The author has revised the paper based on my comments.

As a result, the paper has been improved.

However, about my comments 3 and 5, the revisions are inadequate.

The author states in lines 196-197 of the revised manuscript.

“4. Essential Oil-Based Care in Palliative Care: Mechanisms, Applications, and Clinical Integration”

Please insert the following phrase where it is most appropriate in this area.

“We were motivated by the lack of review articles on Essential Oil-Based Care in Palliative Care."

Author Response

All changes are highlighted in pink.

Comment 1: However, about my comments 3 and 5, the revisions are inadequate.

Response 1: Thank you for your continued feedback. Regarding Comment 3, we would like to clarify that the changes made were in response to overlapping suggestions from another reviewer. To maintain alignment across the feedback received, we focused revisions on contextual background and emphasized pruritus mechanisms, while keeping the primary focus of the review on essential oils. We hope this helps the reader better appreciate the significance of essential oil-based interventions within the broader context of palliative care.

Regarding Comment 5, we have revised the structure as you suggested, combining and clarifying the main body sections. We have also updated the title accordingly, in line with your Comment 7, to reflect the paper's scope and focus better. Since this section represents the central theme of the manuscript, we believe summarizing it further would dilute its intended depth and emphasis. We appreciate your understanding on this point.

Comment 2: 
The author states in lines 196-197 of the revised manuscript. “4. Essential Oil-Based Care in Palliative Care: Mechanisms, Applications, and Clinical Integration”. Please insert the following phrase where it is most appropriate in this area. “We were motivated by the lack of review articles on Essential Oil-Based Care in Palliative Care."

Response 2: Thank you for your helpful suggestion. We have incorporated the sentence “We were motivated by the lack of review articles on Essential Oil-Based Care in Palliative Care” at the beginning of Section 4, as recommended.

Reviewer 3 Report

Comments and Suggestions for Authors

This revision has significantly improved the manuscript. In my view, this manuscript can be published in the present revised form now.

Author Response

Comment 1: This revision has significantly improved the manuscript. In my view, this manuscript can be published in the present revised form now.

Response 1: Thank you very much for your positive feedback and for recognizing the improvements made in the revision. We are pleased to hear that the revised manuscript meets your expectations and appreciate your recommendation for publication.